# Auto-Combustion Synthesis of $Mn_{1−x}Ag_xCo_2O_4$ Catalysts for Diesel Soot Combustion

**Huanrong Liu [1,\*], Yanhong Chen [1], Dongmin Han [1], Weiwei Ma [1], Xiaodong Dai [1] and Zifeng Yan [2]**

[1]  Shandong Institute of Petroleum and Chemical Technology, Dongying 257061, China
[2]  State Key Laboratory of Heavy Oil Processing, China University of Petroleum, Qingdao 266555, China
[\*]  Correspondence: yidian2829@sdipct.edu.cn; Tel.: +86-0546-7396241

**Abstract:** The increase in diesel consumption has led to the proliferation of soot particles from diesel exhaust, resulting in pollution in the form of smog. To solve this problem, a series of Ag-doped $Mn_{1−x}Ag_xCo_2O_4$ spinel catalysts were successfully prepared using an auto-combustion synthesis method that uses glucose as a fuel. X-ray diffraction and Fourier transform infrared spectroscopy analysis were used to analyse the phase structure of the as-prepared samples. The results reveal that the selected catalysts featured a spinel-type structure. Moreover, the catalytic activity of the catalysts for soot combustion was evaluated by temperature-programmed reaction analysis. The temperature required for soot combustion depended heavily on the Ag concentration in the $Mn_{1−x}Ag_xCo_2O_4$ catalyst. The $Mn_{0.8}Ag_{0.2}Co_2O_4$ catalyst had a superior catalytic activity with a $T_{90}$ of 399 °C and $CO_2$ selectivity of 99.3%.

**Keywords:** soot; $Mn_{1−x}Ag_xCo_2O_4$; Ag; auto-combustion synthesis





## 1. Introduction

Soot particulates produced during diesel combustion are a significant source of urban ambient pollution (PM10 and PM2.5), which causes serious respiratory diseases and environmental problems. Therefore, efficient after-treatment systems, namely diesel particulate filters (DPFs), are required to treat the exhaust gas from diesel. Most of the soot particles in exhaust gas can be trapped by the DPF. However, soot deposits block pipes, weaken the purification ability and decrease the engine efficiency. Additionally, the catalyst employed here can catalyse soot combustion within an exhaust temperature range of 150–400 °C, which is much lower than the temperature required for non-catalytic soot combustion (approximately 500–600 °C). Hence, a catalyst with a high activation for soot combustion at low temperatures is highly desirable. The catalytic combustion of diesel soot is a typical gas-solid-solid reaction. Noble metals (Au, Ag, Pt, and Pd) [1–4], transition metal oxides [5], alkaline metal oxides [6], and mixed metal oxides [7,8] have been studied intensively in recent years. Despite their high catalytic activity for soot removal, conventional noble metal catalysts are underutilised because of their high cost and scarcity. Hence, there is an urgent need to develop a new type of catalyst for diesel engine exhaust purification. Spinel-type oxides have proven to be good candidates for the catalytic combustion of diesel soot owing to their good redox properties, thermal stability, and tuneable catalytic performances [9].

The spinel complex oxide has the chemical formula $AB_2O_4$, where A and B occupy the tetrahedral and octahedral sites, respectively (Figure 1). The transition metal cations can shift between the A and B sites. For example, the A ions and half of the B ions can exist in the octahedral sites, while the remaining B ions are located in the tetrahedral sites. This may result in an inverse spinel structure [10,11]. In the case of the spinel-type complex oxide catalyst, B-site cations are beneficial for diesel soot combustion, and the activity of the catalyst can be adjusted by the doping of B-site ions directly or by changing the valence state of B ions via the doping of A-site ions without changing the primary crystal structure [12,13]. In a previous study [14], our $Mn_{1−x}Ag_xCo_2O_4$ spinel oxide catalysts, prepared using a

sol-gel method, exhibited an excellent catalytic performance for soot combustion, and the catalytic activities of the samples increased when increasing the Ag concentration. The $Mn_{0.6}Ag_{0.4}Co_2O_4$ catalysts demonstrated a better catalytic performance for soot combustion than most active catalysts reported for $Pt/SiO_2$ [3] and 3DOM $Au_{1.25}/LaFeO_3$ [1].

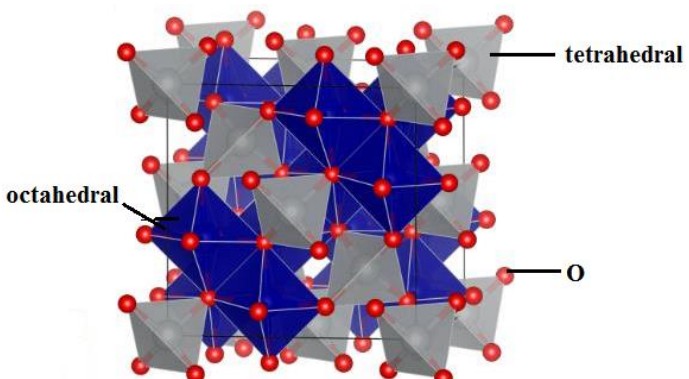

**Figure 1.** Spinel crystal structure.

AB$_2$O$_4$ spinel-type catalysts are typically prepared via annealing at high temperatures of up to 1000 °C or via a complicated approach including dissolution, evaporation, or hydrothermal treatment [15]. In contrast to conventional approaches for preparing spinel-type oxides such as the sol-gel method, precipitation method, and solid-state reaction method, the auto-combustion method promotes the formation of the targeted materials and reduces the roasting temperature via a redox reaction between a metal precursor and organic additives. In our earlier study [16], the auto-combustion method was introduced to prepare a series of $MnCo_2O_4$ spinel-type oxides, and the effects of the synthetic conditions on the structure and soot combustion performance were investigated in detail.

Ag is effective in soot oxidation for the removal of diesel exhaust gas, as demonstrated by a wide range of studies on Ag-supported $MnO_2$-$CeO_2$, $Al_2O_3$ and $LaCoO_3$, or Ag-doped $LaMnO_3$ [17–21]. A series of $Mn_{1-x}Ag_xCo_2O_4$ spinel-type catalysts was prepared by an auto-combustion method in this study. Their physical and chemical properties and the effect of the Ag doping concentration on the structure and catalytic performance were investigated.

## 2. Results and Discussion

### 2.1. X-ray Diffraction (XRD) Results

The XRD patterns of the as-prepared samples are shown in Figure 2. Comparisons with the standard spinel sample (JCPDS 00-023-1237) established that the $Mn_{1-x}Ag_xCo_2O_4$ sample was successfully prepared with a spinel structure through auto-combustion. Meanwhile, relative to an unsubstituted $MnCo_2O_4$ catalyst, the diffraction peak ($2\theta = 38.397°$) corresponding to the (311) plane of the $Mn_{1-x}Ag_xCo_2O_4$ catalyst gradually shifted to a higher $2\theta$ value with an increasing Ag-substitution due to the distortion and expansion of the crystal structure after the doping of Ag into the spinel lattice. Moreover, a trace quantity of Ag impurity phase (JCPDS 01-087-0597) could also be detected when the fraction of Ag was greater than 0.2. This was attributed to the fact that $Ag^+$ could not enter the $MnCo_2O_4$ lattice and form metallic Ag, as the radius of the introduced $Ag^+$ is larger than that of $Mn^{2+}$. In addition, the diffraction intensity decreased gradually at the peak (311), indicating that increasing the Ag content could retard sintering and decrease the particle size in favour of soot combustion.

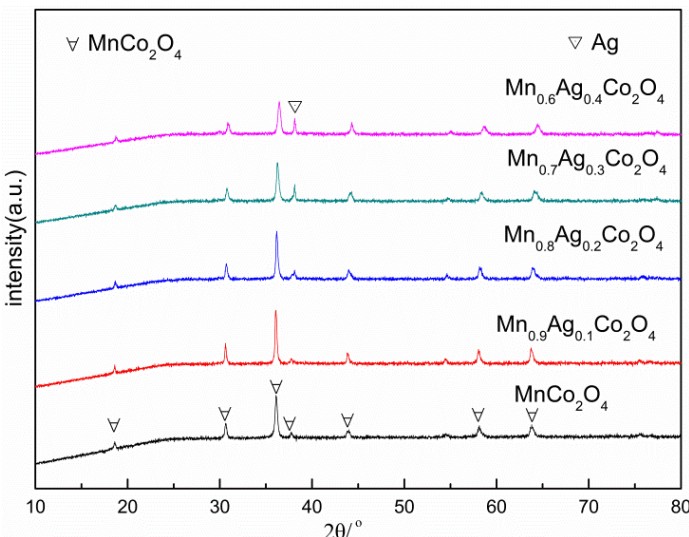

**Figure 2.** X-ray diffraction (XRD) patterns of as-prepared $Mn_{1-x}Ag_xCo_2O_4$ spinel-type samples.

### 2.2. Fourier Transform Infrared (FT-IR) Results

The FT-IR spectra of the as-prepared $Mn_{1-x}Ag_xCo_2O_4$ (x = 0, 0.1, 0.2, 0.3, and 0.4) samples are shown in Figure 3. Two intense vibration peaks appear at approximately 547 cm$^{-1}$ and 649 cm$^{-1}$ in the spectra for every sample. The peak at the lower wavenumber can be ascribed to the stretching vibration of the Co-O bond in the $BO_6$ octahedron, and the peak at the higher wavenumber can be attributed to the stretching vibration of the tetrahedral Mn-O bond in the $AO_4$ tetrahedron [22,23]. The existence of both of these absorption bands is indicative of the formation of a spinel structure. These results are consistent with the XRD results. Moreover, after adding Ag to partially substitute the A-site cations, both vibration peaks shift upfield when increasing the Ag concentration, possibly due to the presence of high-valent $Co^{4+}$, when the $Ag^+$ ions partially replace the $Mn^{2+}$ ions. The wavenumbers of the stretching vibration caused by $Co^{4+}$-O bonding are higher and the interactions are stronger than those caused by $Co^{3+}$-O bonding [24].

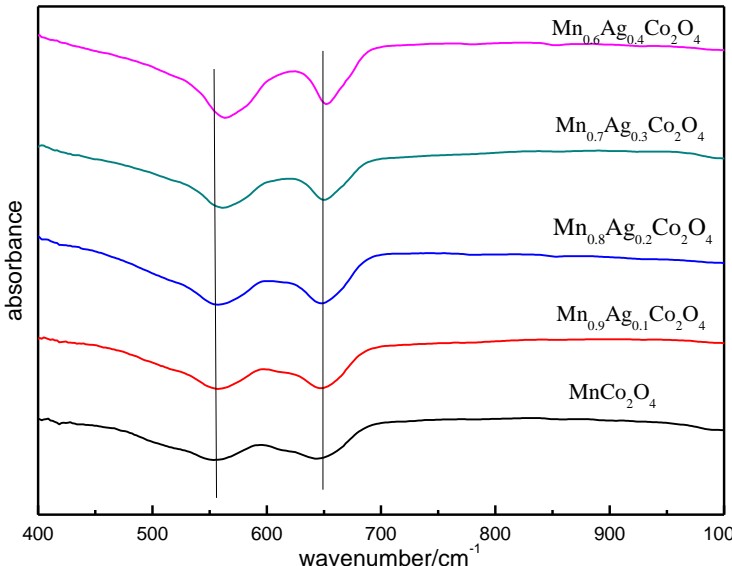

**Figure 3.** Fourier transform infrared (FT-IR) spectra of $Mn_{1-x}Ag_xCo_2O_4$ catalysts.

### 2.3. Temperature-Programmed Reduction with $H_2$ ($H_2$-TPR)

To investigate the reduction behaviour of the catalysts, a $H_2$-TPR analysis was performed, and the results are shown in Figure 4. The TPR spectra reveal two peaks due to the reduction of each cation. As previously reported [25–27], the first peak, observed in the range of 270–410 °C, is attributable to the reduction of Co(III) to metallic Co. Moreover, the peaks are split, which was ascribed to the reduction of $Co^{3+}$ to $Co^{2+}$ (lower temperature) and $Co^{2+}$ to metallic Co (higher temperature). The second peak, which appears at 450–610 °C, was attributed to the reduction of Mn(III) to Mn(II). This peak appears at a lower temperature than that for pure $Mn_2O_3$, owing to the presence of cobalt cations, which are reduced at lower temperatures than Mn. This effect has been previously reported in the literature [28]. In addition, the peaks for the $Mn_{1-x}Ag_xCo_2O_4$ sample shifted to a lower temperature, suggesting that the presence of Ag increased the reducibility of surface oxygen atoms and enhanced oxygen mobility throughout the reduction process [29]. Moreover, the $H_2$ consumption was the highest for the samples catalysed with $Mn_{0.8}Ag_{0.2}Co_2O_4$, which likely led to more surface oxygen being adsorbed through the formed vacancies and structural distortions on the mixed oxide that resulted from the different radii and charges of the $Ag^+$ and $Mn^{2+}$ species.

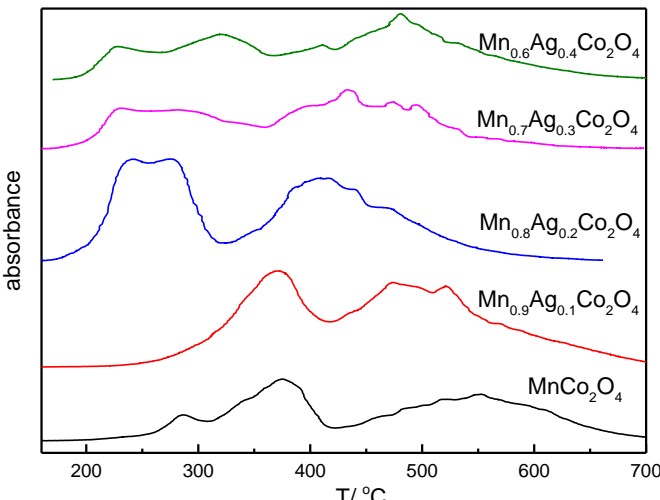

**Figure 4.** $H_2$ temperature-programmed reduction ($H_2$-TPR) profiles of $Mn_{1-x}Ag_xCo_2O_4$ spinel-type catalysts.

### 2.4. Catalytic Activity

Figure 5 shows the catalytic performance of diesel soot oxidation evaluated by temperature-programmed oxidation (TPO) experiments in an $O_2$ atmosphere balanced by He. When compared with pure soot combustion, the results indicate that all catalysts promoted soot combustion. The $T_{10}$, which is the temperature at which 10% of the soot was oxidised during the TPO procedure, decreased from 614 to 400 °C for soot combustion over the $MnCo_2O_4$ catalyst. After Ag doping, the catalytic activity for soot combustion improved markedly. Most significantly, $Mn_{0.8}Ag_{0.2}Co_2O_4$ exhibited the lowest $T_{10}$ (327 °C) and $T_{90}$ (442 °C), reflecting the desirable redox properties of $Mn_{1-x}Ag_xCo_2O_4$. These results are consistent with the TPR results.

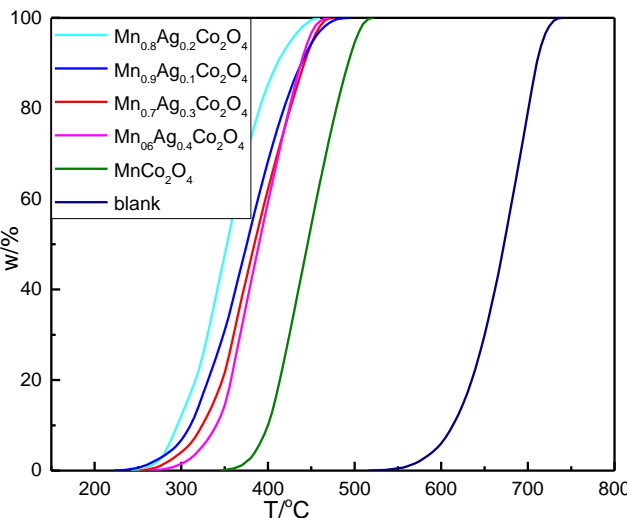

**Figure 5.** Temperature-programmed oxidation (TPO) profiles of soot oxidation over $Mn_{1-x}Ag_xCo_2O_4$ in $O_2$/He atmosphere.

To clarify the role of $NO_x$ in the reaction, we also investigated the catalytic activity of the as-prepared catalysts in a 2000 ppm NO and 5% $O_2$ atmosphere. Figure 6 shows the catalytic performance for soot combustion under these conditions. After introducing 2000 ppm NO to the flow of 5% $O_2$ and He, the catalytic performance for soot combustion improved, obtaining a lower $T_{10}$ and $T_{50}$ than those obtained in an $O_2$ atmosphere (Table 1), which indicates that NO oxidation is a key influence on catalytic soot combustion. Similar conclusions have been reported in the literature [30–33]. The $T_{10}$ (301 °C) and $T_{50}$ (370 °C) of the $Mn_{0.8}Ag_{0.2}Co_2O_4$ catalyst were the lowest among the catalysts tested. The $S_{CO2}$ ($CO_2$ selectivity during the TPRe run) of this catalyst was 99.3%.

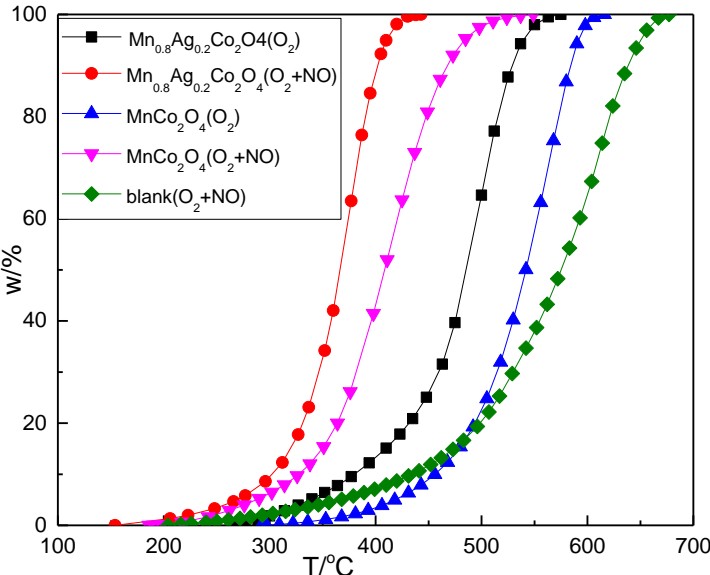

**Figure 6.** Soot catalytic combustion activity of the as-prepared catalyst under different conditions.

**Table 1.** The catalytic activity for soot combustion under the condition of loose contact between catalyst and soot.

| Reaction Conditions | Catalyst | $T_{10}$/°C | $T_{50}$/°C | $T_{90}$/°C | $S_{CO2}$/% | $E_a(T_p)$/kJ/mol |
|---|---|---|---|---|---|---|
| | Blank | 441 | 579 | 641 | 55.0 | 173.6 |
| | $MnCo_2O_4$ | 328 | 409 | 470 | 97.8 | 130.9 |
| | $Mn_{0.8}Ag_{0.2}Co_2O_4$ | 301 | 370 | 399 | 99.3 | 117.5 |
| $O_2$ + NO | 3DOM $Au_{1.25}/LaFeO_3$ | 312 | 387 | 415 | 99.4 | – |
| | $Co_3O_4$ nanowires | 397 | 466 | – | – | – |
| | $Ag/LaCoO_3$-400 | 311 | 363 | 397 | – | 131.7 |
| | $MnCo_2O_4$ | 400 | 446 | 493 | 97.5 | 165.1 |
| $O_2$ | $Mn_{0.8}Ag_{0.2}Co_2O_4$ | 327 | 388 | 442 | 99.0 | 141.4 |
| | $Ag/LaCoO_3$-400 | 392 | 439 | 464 | | 146.7 |

The regenerating lattice vacancies introduced via the doped Ag are extremely important as they adsorb oxygen species and $O^{2-}$ species and promote the combustion of soot particles. The $T_{10}$, $T_{50}$, and $T_{90}$ of $Mn_{0.8}Ag_{0.2}Co_2O_4$ decreased by 140 °C, 209 °C, and 242 °C, respectively, relative to pure soot combustion. $Mn_{0.8}Ag_{0.2}Co_2O_4$ has a better soot combustion activity than the reported catalysts $Co_3O_4$ [5], $Ag/LaCoO_3$-400 [20], and 3DOM $Au_{1.25}/LaFeO_3$ [1], even though the 3DOM $Au_{1.25}/LaFeO_3$ catalyst has a larger surface area and pore size, which improve soot combustion.

The activation energy is an intrinsic property for determining the catalytic activity, and for soot combustion it can be calculated through the so-called Redhead method, using the temperature at which the maximum rate occurs ($T_m$) during TPO experiments [34]. Table 1 shows the $E_a$ values of the catalysts.

The $E_a$ value for $MnCo_2O_4$ under $O_2$ is 165.1 kJ/mol, which is similar to that for non-catalytic soot combustion. The $E_a$ value for $Mn_{0.8}Ag_{0.2}Co_2O_4$ under $O_2$ + NO flow decreased by 56.1 kJ/mol relative to that for non-catalytic soot combustion (Table 1) and decreased by 13.4 kJ/mol relative to that for $MnCo_2O_4$. Ag doping and the introduction of NO reduce the activation energy for soot oxidation because the $NO_2$ formed via the oxidation of NO with surface oxygen species provides an oxidant for soot combustion, particularly since these species have a higher soot oxidation activity than NO and $O_2$ [20].

In summary, based on the above interpretations, $Mn_{1-x}Ag_xCo_2O_4$ catalysts could be successfully synthesised by auto-combustion, and among the catalysts tested, $Mn_{0.8}Ag_{0.2}Co_2O_4$ had the highest catalytic activity for diesel soot combustion. The possible reasons for this are as follows: (1) after lower-valence $Ag^+$ replaces $Mn^{2+}$, for the neutral balance, the valence state of some cobalt ions rises; that is, some of the $Co^{3+}$ ions are oxidized to $Co^{4+}$; (2) the doping of low-valent ions generates many oxygen vacancies, which results in the adsorption of oxygen species and $O^{2-}$ species and promotes the combustion of soot particles [29]; and (3) $Ag^+$, which does not enter the lattice, forms metallic Ag, which collects atomic O and transforms it into highly reactive oxygen, which further catalyses the combustion of soot particles, even though the activity of Ag is lower than that of $Mn_{1-x}Ag_xCo_2O_4$. Overall, the synergy between the Ag species and oxygen vacancies improves soot oxidation [20,35].

### 3. Conclusions

$Mn_{1-x}Ag_xCo_2O_4$ spinel oxide catalysts were synthesised by an auto-combustion method. The as-prepared catalysts demonstrated an excellent catalytic soot oxidation performance compared with bare soot combustion. After the introduction of Ag, the catalytic activity for soot oxidation was further improved. Considering the combined characterisation results and the catalytic performance, the $Mn_{0.8}Ag_{0.2}Co_2O_4$ catalyst was the best candidate [$T_{10}$ (301 °C), $T_{90}$ (399 °C), and $S_{CO2}$ (99.3%)] among the tested catalysts as a result of the formation of oxygen vacancies and the doping of metallic Ag. However, the contact area between the catalyst and solid reactant is the key factor for the reaction, since

the combustion occurs at the three-phase boundary of the soot, catalysts, and reactant gas. Because the catalyst is a non-porous or a microporous nano-powder, it is difficult for soot particles with particle sizes ranging from 20 nm to 1 mm to enter its inner pores [36]. Hence, future studies should focus on the development of catalysts with a three-dimensional, ordered, macroporous structure to improve activity.

## 4. Materials and Methods

### 4.1. Synthesis of Catalysts

Mn(AC)$_2$·4H$_2$O, AgNO$_3$, and Co(NO$_3$)$_2$·6H$_2$O, with a certain molar ratio, were first ground for 15 min. Subsequently, glucose (0.6 times the total metal ion concentration) was added as a fuel, followed by another 25 min of grinding to ensure that the reaction proceeded homogenously. The obtained sample was transferred to an oven at 160 °C for 12 h, and the auto-combustion reaction proceeded slowly. During combustion, gas evolved, and a grey fluffy powder formed in the beaker. Finally, the precursor was annealed for 4 h in a muffle furnace at 700 °C.

### 4.2. Catalyst Characterisation

To examine the crystal structures of the samples, XRD was conducted (Philips PANalytical, X'pert Pro. MPD, Almelo, The Netherlands) using Cu-K$\alpha$ radiation at a scan speed of 2°/min and a 2$\theta$ range of 10–80°. The patterns were compared with JCPDS reference data for phase identification.

FT-IR absorbance spectra were obtained on a Nicolet 6700 spectrometer (The Thermoelectric Company of America, Saddle Brook, NJ, USA) from 4000 to 400 cm$^{-1}$. The sample was prepared as a KBr pellet with a sample-to-KBr-weight ratio of 1:100.

To fully explore the redox ability of the catalyst, H$_2$-TPR analyses were performed on Quantachrome ChemBET-3000 (Boynton Beach, FL, USA). Typically, the sample was pretreated at 300 °C for 1 h in a He flow and then cooled to room temperature. A H$_2$/Ar flow (9.87 vol%, 60 mL/min) was subsequently applied to the heated sample (room temperature to 800 °C). The effluent gas signal was monitored by a thermal conductivity detector.

### 4.3. Activity Evaluation

The performance of the prepared samples was evaluated using a temperature-programmed reaction (TPRe) technique on a fixed-bed. The reaction temperature varied during each TPRe run from 100 °C to 800 °C. The model soot used in this work, Printex-U (Degussa, Frankfurt, Germany), had a primary particle size of 25 nm and specific surface area of 100 m$^2$/g. The catalyst (150 mg) and soot (15 mg) were carefully mixed to simulate the loose contact mode, which is the model most representative of diesel particles flowing through a catalytic filter. Gas containing 5 vol% O$_2$ in Ar was employed. The outlet gases were analysed using gas chromatography. The catalytic performances of the catalysts were evaluated by obtaining the $T_{10}$, $T_{50}$, and $T_{90}$.

The activation energy for soot combustion was determined by the Redhead method using the following equation:

$$k_0 e^{-\frac{E_a}{RT_p}} = \frac{\alpha E_a}{RT_P^2} \tag{1}$$

where $k_0$ is the pre-exponential factor, $E_a$ is the activation energy in kJ/mol, $T_p$ is the temperature at which the maximum rate occurs, and $\alpha$ is the heating rate.

**Author Contributions:** Writing—H.L. and Y.C.; preparation—D.H. and W.M.; investigation—X.D. and Z.Y. All authors have read and agreed to the published version of the manuscript.

**Funding:** This research received no external funding.

**Data Availability Statement:** The results in this manuscript were presented clearly, honestly, and without fabrication, falsification, or inappropriate data manipulation (including image-based manipulation).

**Conflicts of Interest:** The authors declare no conflict of interest.

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
