# Peer review of "Auto-Combustion Synthesis of Mn1−xAgxCo2O4 Catalysts for Diesel Soot Combustion"

_catalysts, doi:10.3390/catal12101182_

Round 1
Reviewer 1 Report
After reading the manuscript, I regard it a good paper worthy of publication in this journal. I only have several remarks to suggest:
1º Literature: All the authors are to be added in the references. They are to be fully acknowledged as well.
2º There is some mistake in naming the figures. Please, revise them.
3º There is certain mess throughout the paper about whether Ag enter or not inside the spinel. I think the authors want to say that Ag enter up to a 0.2 ratio and above it, it is left outside. However, this changes from one point to another of the paper and therefore, it needs to be checked and corrected.
Author Response
Point1 : Literature: All the authors are to be added in the references. They are to be fully acknowledged as well.
Response 1: We appreciated the your suggestions. We have carefully revised the manuscript thoroughly, and updated the text. It is hopeful the revision can improve the paper’s quality and meet the standard requirement
Point 2 : There is some mistake in naming the figures. Please, revise them.
Response 2: The authors revised mistakes and updated the manuscript.
Point 3: There is certain mess throughout the paper about whether Ag enter or not inside the spinel. I think the authors want to say that Ag enter up to a 0.2 ratio and above it, it is left outside. However, this changes from one point to another of the paper and therefore, it needs to be checked and corrected.
Response 3: We agree it is crucial to provide structure-activity relationship for the studied catalysts here. Currently, we apologized that we could not provide the results of this analysis (e.g.SEM-EDX) results due to the incapacity of instrumental analysis. We will continue to work on characterize catalyst characterization.

Reviewer 2 Report
This manuscript describes a study in which the preparation method and silver content of a spinel-type material are shown to be critical to its catalytic activity in soot oxidation. The results are clearly presented and convincingly interpreted on an empirical level.
However, there are some important details missing, which need to be included before the manuscript is suitable for publication in the special issue on synthesis and application of catalytic materials:
In the Introduction, could the authors include brief descriptions of the catalyst technology currently used to control the release of soot particulate from diesel engines, and could they explain why there is still a need for new catalysts? It would also be appropriate to comment on the relative roles of NO2 and O2 in soot oxidation.
In line 59, the authors state that it is well known that silver is a good catalyst for controlling diesel exhaust but, as far as I know, silver is not currently used in any commercial diesel catalyst system. They need to explain that they are referring to early stage research and back up their statement with references.
In the Results and Discussion, Ea values are expressed to 1 decimal place. Is this precision justified? What is the degree of uncertainty in these values? What is the significance of Tp (in brackets after Ea) in the heading of the last column of Table 1?
The caption to Table 1 should make it clear that the three reference catalysts have not been tested in this study, but their performance values have been obtained from literature. It should also be made clear that Ea values are not available for the three reference catalysts.
What is the function of the glucose used during catalyst preparation? Is it simply to generate a large exotherm or does it also produce a reducing atmosphere?
I think readers with a background in surface science will be familiar with the use of the Redhead equation to calculate Ea values for desorption of molecules from a surface, but most readers will not be familiar with its use to calculate the activation energy for a catalytic reaction. It would be very helpful if, in Materials and Methods, the authors can provide the exact form of the Redhead equation that they have used, and if they can explain the assumptions on which their calculations are based. They should also make it clear that the Ea values calculated in this way correspond to apparent activation energies (based on rates of reaction) and not true activation energies (based on rate constants).
In the Conclusions, the authors mention that the activity of their best catalyst still needs to be improved. To help readers appreciate this point, it would be worthwhile comparing the T10 and T90 temperatures with the typical temperatures in a diesel exhaust system.
Finally, it would be very interesting to know whether the trend in relative activity of the catalysts for soot oxidation is the same as the trend for NO oxidation to NO2. This could indicate whether the soot is being oxidised by activated oxygen or by NO2, when both NO and O2 are present in the gas feed. It is worth pointing out that, once it has been formed, NO2 can react directly with soot particles without the need of a catalyst surface – This is the principle behind the commercial catalyst system known as CRT (continuously regenerating trap), in which the function of the catalyst is to generate NO2, which then oxidises the soot that has been trapped in a filter.
Author Response
Point 1: In the Introduction, could the authors include brief descriptions of the catalyst technology currently used to control the release of soot particulate from diesel engines, and could they explain why there is still a need for new catalysts? It would also be appropriate to comment on the relative roles of NO2 and O2 in soot oxidation.
Response 1: We appreciated the your suggestions. We have carefully revised the manuscript thoroughly, and updated the text. It is hopeful the revision can improve the paper’s quality and meet the standard requirement. Details are also shown below:
Beside, the catalyst employed here can catalyze soot combustion within the exhaust temperature range (150-400 oC), extremely lower than the temperature required for non-catalytic soot combustion (ca. 500-600 oC). Hence, a catalyst with high activation for soot combustion at low temperatures is highly desirable. The catalytic combustion of diesel soot is a typical gas-solid-solid reaction. Noble metals (Au, Ag, Pt, and Pd) [1-4], transition metal oxides [5], alkaline metal oxides [6], and mixed metal oxides [7-8] have been studied intensively in recent years. Conventional noble metal catalysts, although with high catalytic activity for the removal of soot, are underutilized because of high price and scarce resource. So the new type of catalyst for diesel engine exhaust purification is urgently needed to be developed. Spinel-type oxides have proven good candidates for the catalytic combustion of diesel soot owing to their good redox properties, thermal stability, and tuneable catalytic performances [9].
As for the roles of NO2 and O2 in soot oxidation, detailed statement has been made In line 167-170.
Point 2: In line 59, the authors state that it is well known that silver is a good catalyst for controlling diesel exhaust but, as far as I know, silver is not currently used in any commercial diesel catalyst system. They need to explain that they are referring to early stage research and back up their statement with references.
Response 2: Ag active components has been used in many industrial application such epoxidation of ethylene to ethylene epoxide [1], as well as selective catalytic reduction for NOx abatement [2]. In the field of soot oxidation, silver and/or silver oxide (Ag2O) has been recognized as the active sites or components as individual compounds or active species on support (CeO2, ZrO2, Al2O3, etc)[3-6].
[1] Yukihide Shiraishi,Naoki Toshima. Colloidal silver catalysts for oxidation of ethylene.Journal of Molecular Catalysis A: Chemical, 1999, 141: 187-192.
[2] P. P W, B. C L. Effect of SO2 on the activity of Ag/γ-Al2O3 catalysts for NOx reduction in lean conditions. Applied Catalysis B: Environmental 2005,59(1-2): 27-34.
[3]Villani K, Brosius R, Martens J A. Catalytic carbon oxidation over Ag/Al2O3. Journal of Catalysis, 2005, 236(1): 172-175.
[4]Aneggi E, Llorca J, de Leitenburg C, et al. Soot combustion over silver-supported catalysts. Applied Catalysis B: Environmental, 2009, 91(1): 489-498.
[5]Gardini D, Christensen J M, Damsgaard C D, et al. Visualizing the mobility of silver during catalytic soot oxidation. Applied Catalysis B: Environmental, 2016, 183: 28-36.
[6]Shimizu K, Kawachi H, Satsuma A. Study of active sites and mechanism for soot oxidation by silver-loaded ceria catalyst. Applied Catalysis B: Environmental, 2010, 96(1): 169-175.
The authors updated the manuscript as follows:
In soot oxidation for removal of diesel exhaust gas, Ag supported MnO2-CeO2, Al2O3, LaCoO3, and Ag doped LaMnO3 have been wildly reported, disclosing Ag is effective in soot oxidation [17-21].
The authors also updated the literature as follows:
[17]Eun Jun Lee,Min June Kim,Jin Woo Choung,et.al. NOx-assisted soot oxidation based on Ag/MnOx-CeO2 mixed oxides. Applied Catalysis A: General 2021,627:118396-118405.
[18]Zhen Zhao, Jing Ma, Min Li, et.al. Model Ag/CeO2 catalysts for soot combustion: Roles of silver species and catalyst stability. Chemical Engineering Journal 2022,430: 132802-132812.
[19]Villani K, Brosius R, Martens J A. Catalytic carbon oxidation over Ag/Al2O3. Journal of Catalysis 2005, 236(1): 172-175.
[20]Qi Fan, Shuai Zhang, Liying Sun, et.al. Catalytic oxidation of diesel soot particulates over Ag/LaCoO3 perovskite oxides in air and NOx. Chinese Journal of Catalysis 2016,37(3):428-435.
[21]Wang Ke, Liu Huanrong, Yan Zifeng. Simultaneous removal of NOx and soot particulates over La0.7Ag0.3MnO3 perovskite oxide catalysts. Catalysis Today 2010,158(3-4): 423-426.
Point 3: In the Results and Discussion, Ea values are expressed to 1 decimal place. Is this recision justified? What is the degree of uncertainty in these values? What is the significance of Tp (in brackets after Ea) in the heading of the last column of Table 1?
Response 3: Activation energies are needed for rational designs of soot combustion reaction. Hence the activation energy for soot oxidation was determined for the catalyst by the Redhead method, using peak temperature data from TPO experiments.[ B. Dernaika, D. Uner. A simplified approach to determine the activation energies of uncatalyzed and catalyzed combustion of soot. Applied Catalysis B: Environmental 2003, 40 : 219–229.]
Text updated the specific calculation formula as follows:
The activation energy of the soot combustion is determined by the Redhead method [38] using the following expression:
k0e−Ea/(RTp) =αEa/(RTp2) (1)
where k0 is pre-exponential factor, E is activation energy in kJ/mol, Tp is the temperature at which maximum rate occurs, α is the heating rate employed.
[38] B. Dernaika, D. Uner. A simplified approach to determine the activation energies of uncatalyzed and catalyzed combustion of soot. Applied Catalysis B: Environmental 2003, 40 : 219–229.
Point 4: The caption to Table 1 should make it clear that the three reference catalysts have not been tested in this study, but their performance values have been obtained from literature. It should also be made clear that Ea values are not available for the three reference catalysts.
Response 4: In the research field of catalytic soot oxidation, The catalyst activity is evaluated by T10, T50, T90 or Ea, In the reference, only T10, T50 and T90 are listed as a standard for evaluating catalysts 3DOM Au1.25/LaFeO3 and Co3O4 nanowires, that T10, T50, T90 and Ea for evaluating catalysts Ag/LaCoO3-400. These are to be consistent with the literature data.
Point 5: What is the function of the glucose used during catalyst preparation? Is it simply to generate a large exotherm or does it also produce a reducing atmosphere?
Response 5: The auto-combustion with the additive of fuel, can facilitate the formation of mixed oxides via a highly exothermic redox reaction between oxidants (metal precursors) and reducing agents (organic additive), with low-agglomeration features and a simple preparation. In terms of this approach, the introduced fuel, glucose plays a crucial role in the microstructure including surface area, crystal structure of prepared materials, and this approach can prepare the targeted materials via a low temperature[1-6].
[1]A.B.Salunkhe,V.M.Khot,M.R.Phadatare,N.D. Thorat.Low temperature combustion synthesis and magnetostructural properties of Co–Mn nanoferrites. Journal of Magnetism and Magnetic Materials 2014, 352: 91–98.
[2]A.B. Salunkhe, V.M. Khot, M.R. Phadatare, S.H. Pawar. Combustion synthesis of cobalt ferrite nanoparticles Influence of fuel to oxidizer ratio. Journal of Alloys and Compounds 2012, 514: 91-96.
[3]G. Jaya Rao R. Mazumder , S. Bhattacharyya, P. Chaudhuri. Synthesis CO2 absorption property and densification of Li4SiO4 powder by glycine-nitrate solution combustion method and its comparison with solid state method. Journal of Alloys and Compounds 2017, 725: 461-471.
[4]A.Varma, A.S.Mukasyan, A.S.Rogachev, K.V.Manukyan. Solution combustion synthesis of nanoscale material. Chem.Rev 2016,116(23): 14493-14586.
[5]Vijay Singh, N. Singh, M.S. Pathak, Vikas Dubey, Pramod K. Singh. Annealing effects on the luminescence properties of Ce doped ZnAl2O4 produced by combustion synthesis 2018, 155:285–291.
[6]Robert IanoÅŸ, Roxana Băbuţă. Combustion synthesis of ZnAl2O4 powders with tuned surface area. Ceramics International 2017,43:8975–8981.
Point 6: I think readers with a background in surface science will be familiar with the use of the Redhead equation to calculate Ea values for desorption of molecules from a surface, but most readers will not be familiar with its use to calculate the activation energy for a catalytic reaction. It would be very helpful if, in Materials and Methods, the authors can provide the exact form of the Redhead equation that they have used, and if they can explain the assumptions on which their calculations are based. They should also make it clear that the Ea values calculated in this way correspond to apparent activation energies (based on rates of reaction) and not true activation energies (based on rate constants).
Response 6: The authors appreciated the editor’s suggestions. The author have carefully revised the manuscript thoroughly, and updated the text as follows:
The activation energy of the soot combustion is determined by the Redhead method [38] using the following expression:
k0e−Ea/(RTp) =αEa/(RTp2)
where k0 is pre-exponential factor, E is activation energy in kJ/mol, Tp is the temperature at which maximum rate occurs, α is the heating rate employed.
Point 7: In the Conclusions, the authors mention that the activity of their best catalyst still needs to be improved. To help readers appreciate this point, it would be worthwhile comparing the T10 and T90 temperatures with the typical temperatures in a diesel exhaust system.
Response 7: The authors updated the manuscript as follows:
However, the contact area between the catalyst and solid reactant is the key of reaction since the combustion reaction occurred at the three phase boundary of soot, catalysts and reactant gas. The catalyst is a non-porous or micropore nano powder, soot particles with particle size ranging from 20 nm to 1 mm are difficult to enter the inner pores of catalysts[37].
[37] J.P.A. Neeft, O.P. van Pruissen, M. Makkee, et.al.Catalysts for the oxidation of soot from diesel exhaust gases II. Contact between soot and catalyst under practical conditions. Applied Catalysis B:Environmental 1997, 12:21-31.
Point 8: Finally, it would be very interesting to know whether the trend in relative activity of the catalysts for soot oxidation is the same as the trend for NO oxidation to NO2. This could indicate whether the soot is being oxidised by activated oxygen or by NO2, when both NO and O2 are present in the gas feed. It is worth pointing out that, once it has been formed, NO2 can react directly with soot particles without the need of a catalyst surface – This is the principle behind the commercial catalyst system known as CRT (continuously regenerating trap), in which the function of the catalyst is to generate NO2, which then oxidises the soot that has been trapped in a filter.
Response 8: The problem is related to the catalytic mechanism. The reaction mechanism is different with different reaction atmosphere. Main tentative pathways for the simultaneous reduction of NO and combustion of soot over nanostructured spinel-type oxides catalyst are as follows:(Legend: Oad, NOad= oxygen, nitrogen oxide chemisorbed over the catalyst; Cf= active carbon site; C[i, j ] = i, j species adsorbed over carbon; g = gas phase)
In O2 atmosphere [1]
O2(g) ↔ 2Oad
Cf+ Oad→ C[O]
C[O] + Oad→ CO2(g)(+Cf)
C[O] + (1/2)O→ CO2(g)(+Cf)
C[O] → CO(g)(+Cf)
In NO+O2 atmosphere[2–5]
NO (g) ↔ NOad
Cf+ NOad→ C[N,O]
C[N,O]+ NOad→ O2(g)+N2(g) (+Cf)
C[N,O]+ NO(g)→ CO2(g) + N2(g)
C[N,O]+ NOad (+Cf) → CO2(g) +C[N, N]
C[N, N] → C + N2(g)
NO(g) +1/2O2→ NO2
NO2(g) ↔ NO(ad)+O(ad)
NO2(g)+C→ CO(g) + NO(g)
C[N, N] +Oad→ C + N2O(g)
N2O→Oad+ N2
N2O+Oad→O2+ N2
So soot catalytic oxidation reaction is a complex gas-solid reaction,It is difficult to indicate which step of the reaction the catalyst plays in the reaction.
[1] K. Yoshida, S. Makino, S. Sumiya, G. Muramatsu, R. Helferich, SAE Paper No. 892046 (1989).
[2] K. Matsuoka, H. Orikasa, Y. Itoh, P. Chambrion, A. Tomita. Reaction of NO with soot over Pt-loaded catalyst in the presence of oxygen Appl. Catal.B 2000,26 ,89-99.
[3] Y. Teraoka, K. Nakano, S. Kagawa, W.F. Shangguan. Simultaneous removal of nitrogen oxides and diesel soot particulates catalyzed by perovskite-type oxides. Appl. Catal. B 1995,5:L181-L192.
[4] W.F. Shangguan, Y. Teraoka, S. Kagawa. Simultaneous catalytic removal of NOχ and diesel soot particulates over ternary AB2O4 spinel-type oxides.Appl. Catal. B 1996,8:217-227
[5] Y. Teraoka, K. Nakano, W.F. Shangguan, S. Kagawa. Simultaneous catalytic removal of nitrogen oxides and diesel soot particulate over perovskite-related oxides. Catal. Today 1996,27: 107-118
